# Sustainability in Aquaponics: Industrial Spirulina Waste as a Biofertilizer for *Lactuca sativa* L. Plants

**DOI:** 10.3390/plants12234030

**Published:** 2023-11-30

**Authors:** Davide Frassine, Roberto Braglia, Francesco Scuderi, Enrico Luigi Redi, Angelo Gismondi, Gabriele Di Marco, Lorenza Rugnini, Antonella Canini

**Affiliations:** 1Department of Biology, Tor Vergata University of Rome, Via della Ricerca Scientifica 1, 00133 Rome, Italy; davide.frassine91@gmail.com (D.F.); roberto.braglia@uniroma2.it (R.B.); francesco.scuderi@uniroma2.it (F.S.); enrico.luigi.redi@uniroma2.it (E.L.R.); gismondi@scienze.uniroma2.it (A.G.); gabriele.di.marco@uniroma2.it (G.D.M.); lorenza.rugnini@uniroma2.it (L.R.); 2PhD Program in Evolutionary Biology and Ecology, Tor Vergata University of Rome, Via della Ricerca Scientifica 1, 00133 Rome, Italy

**Keywords:** soilless systems, lettuce, foliar spray fertilizer, spirulina, circular economy, agriculture sustainability

## Abstract

Aquaponics represents an alternative to traditional soil cultivation. To solve the problem of nutrient depletion that occurs in this biotechnological system, the application of a spirulina-based biofertilizer was assessed. The microalgal waste used in this study came from industrial processing. Four different dilutions of the supernatant portion of this waste were sprayed on lettuce plants cultivated in an aquaponics system installed at the Botanical Gardens of the Tor Vergata University of Rome. The biofertilizer was characterized to evaluate its amount of macro- and micronutrients. The analysis conducted on the plants involved both morpho-biometric aspects and qualitative–quantitative measurements. The experiments showed that the spirulina extract had a positive effect on the growth and nutraceutical content of the lettuce plants; the obtained results highlighted that a dilution of 75% was the best for treatment. The use of the proposed organic and recycled fertilizer could increase the sustainability of crop cultivation and promote the functioning of aquaponics systems.

## 1. Introduction

According to a recent estimate, the world’s population has reached 8 billion inhabitants [1]. This number is alarming when considering the global need for food and how this food is, unfortunately, not distributed equally across the world’s various countries; around 30% of the world population (2.37 billion people) do not have access to adequate amounts of food [2]. Mankind of the 21st century must face the problems relating to the ever-increasing number of people and climate change, adopting new strategies for a more sustainable future compatible with the related food needs. Traditional farming systems are, in these terms, no longer sustainable. These methods involve and/or are linked to various problems, such as excessive use of freshwater reserves, climate change, use of pesticides and fertilizers, and contamination with heavy metals, which negatively impact the surrounding environment, reducing its biodiversity, inducing fragmentation of its habitats, and altering its soil composition. Sustainable agriculture refers to the creation of a cultivation system that aims for the reduction of environmental impacts and waste and the conservation and protection of the Earth’s resources; it includes promoting the resilience of agricultural systems and their self-regulation, a transformation towards more innovative and low-impact technologies, and changes to the production capacity [3]. In this way, aquaponics can represent an alternative to soil-based agriculture systems. It is well known that agri-food production in aquaponics has numerous advantages, such as saving water, the absence of pesticides, a greater control over crops, a quicker production rate, and being free from seasonality and climate changes [4,5,6]. However, it is also appropriate to mention the negative aspects of this new cultivation method, like its high initial cost, requirement of great amount of expertise, limited choice of plant species, and its not-always optimal concentration of nutrients [7,8]. Its reduced availability of both macro- and micronutrients has been documented in the research on aquaponics cultivation systems [9,10]. Recent studies have focused on balancing the nutrients in aquaponics, working on both the wastewater from the aquaculture unit and on the bacterial community present in the media in order to make the entire system increasingly sustainable and circular [11,12,13,14]. The present study investigates another method to overcome this lack of nutrients and to improve the yield of plants sustainably and in an eco-friendly way, without changing the water parameters and without interfering with fish breeding. In detail, this research consists of evaluating the effects of the application of a foliar fertilizer based on industrial spirulina (*Arthrospira platensis* (Nordstedt) Gomont) waste biomass on the growth and quality of lettuce plants (*Lactuca sativa* L.) cultivated in aquaponics. The analyzed parameters in this study involved morphological data (size and weight), the content of soluble solids, chlorophylls, carotenoids, phenols, and flavonoids. Studies using spirulina (or other biofertilizers) in aquaponics systems are limited in the literature [15,16]. The use of microalgae as fertilizing biocompounds in agriculture represents a topic that is attracting increasing interest; in this way, they are configured as possible solutions with a low ecological impact capable of promoting sustainability in this field [17]. On the other hand, several works have also reported the effects of inorganic substances, both introduced directly into the water and administered via the leaves, on the growth and physiological state of plants grown in aquaponics [18,19]. The current work could potentially constitute a starting point for the application of industrial spirulina waste as a foliar biofertilizer in aquaponics.

## 2. Results

### 2.1. Size Measurements

Our morphometric analyses (Table 1) included the rosette diameter and root length. Overall, all spirulina-based treated groups revealed significant differences when compared to the control group (CT) for the rosette diameter; among them, the plants exposed to 75% treatment showed the greatest growth compared to the 25%, 50%, and 100% groups. Specifically, the recorded dimensions increased for the 75% treatment group by 61.76% compared to the control, and by 16.54% compared to the 100% group. In contrast, the data on the length of the roots revealed no statistically significant differences among the various treatments.

### 2.2. Fresh and Dry Weight

Overall, the fresh weight (FW) measurements and fresh weight/dry weight ratios (FW/DW) (Table 2) showed significant differences among the treatments, except for the data on root fresh weight, which showed no statistical evidence of differences. In particular, regarding the results of the rosette fresh weight, the 50%, 75%, and 100% treatment groups were significantly heavier than the control, presenting increases of 42.90%, 81.98%, and 53.27%, respectively. Furthermore, the 75% treatment group recorded a higher accumulation of biomass than 25% group (+38.29%). For the sum of the rosette and root FW (total FW), the 50%, 75%, and 100% groups evidenced increases in weight (+37.54%, +70.83%, and +52.36%, respectively) when compared to the control. Also, in this case, the 75% administration group resulted in higher values compared to the 25% group (+34.60%). The FW/DW ratio presented differences regarding the 75% and 100% groups when compared to the 50% one.

### 2.3. SSC, Photosynthetic Pigments, and Secondary Metabolites

The data concerning the SSC, photosynthetic pigments, and secondary metabolites showed significant differences among the various treatments (Table 3). SSCs were different between the 75% and 100% groups compared with the control, showing increases of 92.36% and 71.76%, respectively. Furthermore, the SSC had higher values for plants belonging to the 75% group (+38.85%) than those in the 50% one. For the photosynthetic pigments, CHL *a*, CHL *b*, and CAR contents were analyzed. No statistical evidence was found for the CHL *a* data among the groups treated with the spirulina-based fertilizer and the control; the highest difference values were found for the 50% (+25.53%) and 75% (+25.45%) groups when compared to the 100% group. CHL *b* data from the 50% and 75% groups were the only data presenting a significant difference when compared to the control (+331.46% and +326.41%, respectively) and to the 25% group (478.20% and 471.44%, respectively). The same trend was found for the CAR, with higher values for the 50% and 75% groups when compared to the control (+33.19% and +36.38%, respectively) and to the 25% group (+30.85% and +33.99%, respectively). The 75% treatment favored the accumulation of PHE with respect to the control group (+110.94%) and to the 100% treatment group (+202.97%); an increase in the PHE was also observed between the 25% and the 50% treatment groups when compared with the control group (+86.51% and +101.54%, respectively) and between these same treatment groups when compared with the 100% treatment group (+167.88% and +189.47%, respectively). Regarding the FLA, the treatments seemed not to modify the level of these secondary metabolites, except in the 75% treatment group, where the biosynthesis of these compounds appeared to be negatively affected compared to the control (−28.92%) and to the 25% group (−27.08%).

## 3. Discussion

The present study aimed to investigate the effects of industrial spirulina-based waste, when used as a biofertilizer, on the morphological and nutraceutical level of lettuce plants grown in an aquaponics system. As reported in other works, aquaponics systems are efficient for lettuce production [12]. The choice of this microalga chain was the result of two reasons: The first consisted of the richness in organic and inorganic compounds present in *A*. *platensis* that make it a potential excellent fertilizer [20]. The second was linked to the idea of reinforcing the concept of a circular economy of resources, by giving value to a significant industrial waste biomass [21]. Using a fertilizer in the form of a foliar spray is an efficient strategy in aquaponics because it solves the problems that arise regarding the health and the physiological state of the fish and bacteria present in this system when fertilizers are poured into the water. Moreover, this application proves to be easier to implement and manage compared to other studies conducted so far, which have reported the implementation of aquaponics systems with variations related to the operation or structure of the system, as well as the diversity and density of the associated fish and microbial components [12,22,23,24]. The choice of lettuce was guided by several reasons. *L*. *sativa* plants are, in fact, among the most commercialized green crops in the world; they are very suitable for growing in hydroponic systems and have a fairly fast growth cycle [25,26]. In the present work, different dilutions (i.e., 100%, 75%, 50%, and 25%) of the supernatant portion of industrial spirulina waste biomass were tested. By comparing the data related to control plants, it was possible to define the optimal concentration of foliar biofertilizer to be used for improving lettuce yield and nutraceutical content. The morphological and biometrical aspects considered for this study were the diameter of the rosettes, length of the roots, fresh weight of the rosette, fresh weight of the roots, total fresh weight, and the FW/DW ratio. The evidence found in this research showed that the 75% dilution of the biofertilizer had the most significant effect on the growth of the lettuce plants, with the highest values of all of the groups. Data from the literature regarding treatments using spirulina extracts and/or other cyanobacteria/microalgae support the increase in plant biomass recorded in this study [27,28,29]. The morpho-biometrical data obtained from these experiments showed that the spray application of the spirulina-based biofertilizer promoted rosette growth rather than root growth, as confirmed by further studies [30]. This has a strong significance at both a productive and economic level, as the leaves are the only portion that have dietary purposes. Spectrophotometric/refractometric assays used to investigate both the qualitative and quantitative aspects of the lettuce’s soluble solids, chlorophylls, carotenoids, phenols, and flavonoid contents demonstrated that the 75% treatment induced the synthesis of these compounds, except the FLA. These parameters were chosen as they give information on the main characteristics of vegetal species intended for food consumption, both from the plant’s physiological state (stressed or healthy) and from a nutritional/nutraceutical point of view [31,32]. The refractometric analysis of the SSC allowed us to determine a ‘quality’ parameter, mainly concerning the sugar content, of the plant’s vegetables, such as the fruits or leaves [33]. Indeed, the use of a spirulina-based fertilizer had a positive effect on the accumulation of sugars, as one could expect [34,35,36]. For instance, a similar case study has reported that in *Eruca sativa* Mill., the application of a spirulina-based extract increased the amount of sugars, proteins, and free amino acids [37]. In addition, the levels of soluble solids content found in this study were higher than those obtained in other cultivars of *L. sativa* cultivated in a hydroponic system, indicating, as reported by the authors, that this parameter is strongly cultivar-dependent [38]. In this research, lettuce plants treated with the spirulina waste also responded positively regarding their levels of chlorophylls and carotenoids. Even the increase in the content of photosynthetic pigments in this case post-treatment with spirulina-based spray fertilizer finds support in other studies [39,40]. A common result obtained after the application of biofertilizers, or biostimulants, is an increase in the photosynthetic activity, which reflects an increased synthesis of photosynthetic pigments (e.g., chlorophyll *a*, *b*, and carotenoids) [41,42]. Investigations conducted on seaweed extracts have shown that their administration to cultivated plants promoted plant development, leaf expansion, and the amount of photosynthetic pigments, due to the presence of plant growth regulatory substances (e.g., auxins, gibberellins, betaines, etc.) [43,44]. Potentially, given the results obtained in this study, although other specific analyses should be carried out, it is possible to suppose that the biofertilizer here proposed also contains such types of plant hormones performing similar effects. In addition, the high content of macro- and micronutrients (i.e., N, P, K, Fe, Cu, Mn, Mg, S, Mo, Cl, Zn, and Ca) present in the spirulina extract used in this study could justify the promoting effect of the fertilizer on the growth of the lettuce plants, probably increasing the synthesis of photosynthetic pigments. Our analysis of the phenolic content provided further evidence of the efficacy of the industrial spirulina waste-based biofertilizer on the quality of lettuce plants grown in aquaponics. Phenols represent a variegated family of compounds, characterized by the presence of at least one aromatic ring bearing one or more hydroxyl groups; they are ubiquitous in plants and perform numerous functions, with radical scavenging being the most important, which is connected to the antioxidant properties of the plant food products [45]. It has been demonstrated that consuming plants (extracts or portions of them) that are rich in phenolics can have a positive effect on the reduction of the onset of metabolic disorders (e.g., diabetes), cardiovascular diseases, tumors, and neurodegenerative and proinflammatory conditions [46,47]. The data reported in this research allowed us to suppose that the biosynthesis of phenolic compounds increased when plants were treated with the 25%, 50%, and 75% dilutions. Similar results have been observed in *Zea mays* L. after administration of a spirulina-based extract [48]. Current studies on the composition of bioactive compounds have shown that lettuce plants grown in aquaponics systems achieve very good nutritional value, and their composition of phenolic compounds outperforms that of lettuce grown traditionally in a greenhouse [49]. The data relating to the total flavonoid content differed from the trend observed in the other parameters. Therefore, this evidence led us to hypothesize that the industrial spirulina waste had a negative impact on the production of flavonoids, perhaps altering the activity of specific enzymes (e.g., chalcone synthase). Another option is that the plants direct all their energy to tissue growth, reducing the expensive synthesis of FLA. However, at this stage, interpreting these data is difficult and further investigations will surely be needed to clarify this aspect. The results of the documented experiments exhibit significant fluctuations as the growth performance of lettuce is contingent upon diverse elements, including environmental conditions, the species of fish paired with it, the density of the fish store, the planting density, and variables associated with fish feeding and growth; for this reason, comparisons with other data in the literature and the subsequent conclusions that can be drawn, bearing in mind the fact that studies on the functional responses of plants in aquaponics environments are almost absent, are arduous [50].

## 4. Materials and Methods

### 4.1. Aquaponics System

Experiments were carried out from October 2022 to March 2023 in the aquaponics greenhouse installed at the Botanical Gardens of the Tor Vergata University of Rome (Rome, Italy). The aquaponics unit consisted of two 4000-liter fish tanks stocked with tilapias (*Oreochromis niloticus* L.). It was equipped with an ultraviolet light (UV) sterilizer, a reverse osmosis system, a static 5000-liter biofilter filled with bio-media, and a bottom-up oxygenation system. The flow rate in the biofilter was 2.5 m^3^ h^−1^ and the retention time was 0.8 h ± 45 min. The biofilter supplied two 27.5 m^2^ floating raft system units for the vegetables. Tilapias were fed with a 35% protein fish diet. Furthermore, data relating to the parameters constituting the system (i.e., temperature, pH, dissolved oxygen, conductivity, pressure, irradiance) were recorded using specific sensors present in the structure; abiotic data relating to the greenhouse (i.e., minimum–maximum temperature and minimum–maximum relative humidity) were recorded with a thermohydrometer (Table 4). The water characteristics of the aquaponics system were monitored weekly using special spectrophotometric tests (Hanna Instruments, Woonsocket, RI, USA), which made it possible to observe the concentrations of macro- and micronutrients over time (N, P, K, Fe, Cu, Mn, Mg, S, Mo, Cl, Zn, Ca) (Table 5).

### 4.2. Plant Material and Experimental Set-Up

Seeds of *L. sativa* (cv. Foglia di Quercia Verde) were sown directly into growth baskets sat in custom-made holes (35 mm in diameter) created in closed-cell expanded polystyrene floating panels; approximately 50 g of red lapillus was used as an inert substrate. The experiment (Figure 1) involved the use of 15 floating panels, each containing 5 plants.

### 4.3. Preparation of the Treatments Based on Spirulina Waste Biomass

The *Spireat* company (Caselle, Cremona, Italy) kindly supplied us with the frozen spirulina waste (*A. platensis)*. The residual biomass used in this study was obtained after the extraction of anthocyanins and phycocyanins. This process did not involve the use of any type of solvent. Cell disruption was carried out via freezing/thawing. The extraction procedure was carried out in an aqueous saline solution, while purification of the extract was performed following two membrane filtration steps (microfiltration 0.2 µm/ultrafiltration 50 kDa). The frozen microalgal biomass was thawed in a controlled environment at 4 °C and subsequently observed under an optical microscope (Motic Model BA2010, Xiamen, China) at 40× magnification, to check the complete destruction of the cells. Afterwards, the algal material was left to settle for 3 days and then centrifuged at 4500× *g* for 20 min. The residual pellet and supernatant were divided; this study focused only on the liquid portion, which was collected and subjected to chemical characterization. These analyses were carried out using special spectrophotometric assays (model Iris HI801; Hanna Instruments, Woonsocket, RI, USA), allowing us to measure the macro- and micronutrient compositions in the proposed fertilizer; the same procedures were also performed to analyze the water in the aquaponics system (Table 5). The supernatant was tested as absolute (100%) and in three different dilutions made with sterile double-distilled water (75%, 50%, and 25% *v*/*v*). Sterile double-distilled water was used as the control. All treatments were performed as foliar spray fertilizations on the cultivated plants. A total of three administrations were performed for each treatment: on the 22nd day after sowing (1.5 mL for each plant), on the 34th day (2 mL for each plant), and on the 44th day (4 mL for each plant). The days on which the plants were treated were chosen based on the period required to reach the dimensions of a marketable and consumable product (60–70 days), as indicated in the optimal sowing and harvesting instructions on the back of the commercial package (Blumen Vegetable Seeds, Piacenza, Italy). The sprayed volume in the three administrations was chosen by carrying out some tests to ensure the entire coverage of the leaf surface (the lower and upper lamina) at each specific stage of growth. Since different treatments could involve contiguous floating panels, a mobile plexiglass panel was used to guarantee administration to the interested plant only, thus avoiding experimental contamination between treatments.

### 4.4. Plant Sampling, Biomass Evaluation, and Size Measurements

*L. sativa* plants were harvested after 65 days after sowing. Part of the samples was used immediately for both the wet and dry weight and for the size measurements; the remaining fraction was stored at −80 °C for subsequent analyses. Fresh weight (FW) was evaluated using a precision scale, while for the dry weight quantitation, the samples were subjected to a heat treatment using a ventilated oven set at 55 °C for 48 h. Fresh and dry weightings were performed on both the rosettes and roots. Plants were photographed using a Canon EOS 550D, positioning the camera lens at 70 cm from the samples. The acquired pictures were processed with the ImageJ 1.8.0 software (U.S. National Institution of Health, Bethesda, MD, USA) for determining the rosette diameter and root length.

### 4.5. Soluble Solid Content (SSC) Determination

For the determination of the SSC, the protocol reported in Braglia et al. [51] was followed. Briefly, a total of 1.5 g FW of the randomized samples taken from inner, middle, and outer leaves was homogenized with a mortar and pestle, using liquid nitrogen, and centrifuged at 6089× *g* for 10 min. Then, 100 µL of the supernatant was collected and analyzed using a digital refractometer (model HI96800; Hanna, Woonsocket, RI, USA). This type of analysis allowed us to detect the content of sugars and other soluble solids present in the extracts (Brix value).

### 4.6. Photosynthetic Pigments

A total of 500 mg FW of the randomized samples, taken from the inner, middle, and outer leaves, was homogenized with a mortar and pestle, using liquid nitrogen. Pigments were extracted with 1.5 mL of 80% acetone in a controlled environment of 4 °C and for 24 h in the dark. Subsequently, the extract was centrifuged at 4500× *g* for 10 min and the supernatant was collected for the determination of the pigments. Chlorophyll *a* (CHL *a*) and *b* (CHL *b*) and carotenoid (CAR) contents were determined using the equations previously described by Lichtenthaler [52], by measuring the absorbance of the extracts at 663, 644, and 452 nm, respectively, using a spectrophotometer (model Iris HI801; Hanna, Woonsocket, RI, USA). Results were expressed as µg g^−1^ of the FW.

### 4.7. Total Phenolic and Flavonoid Content

A total of 500 mg FW of the randomized samples, taken from the inner, middle, and outer leaves, was homogenized with a mortar and pestle, using liquid nitrogen. Subsequently, the homogenate was extracted overnight with 1.5 mL of methanol 100% (*v*/*v*) for analysis of the total phenolic content (PHE) and with 1.5 mL of methanol 50% (*v*/*v*) for the evaluation of the total flavonoid content (FLA). The extraction process was performed using an orbital shaker at 110 rpm at room temperature for 48 h. Then, the plant samples were centrifuged at 8603× *g* for 20 min and the supernatant was collected. PHE determination was performed as reported in Di Marco et al. [53], using the Folin–Ciocalteu reagent. Specifically, 200 µL of extract was mixed with 1 mL of Folin–Ciocalteu reagent (previously diluted 1:10; *v*/*v*) and 800 µL of 1 M Na_2_CO_3_; the obtained mixture was then incubated for 1 h at room temperature. For FLA quantitation, the protocol reported in Chang et al. [54], using the aluminum chloride method, was followed. In this case, 200 µL of plant extract was added to 40 µL of 10% AlCl_3_, 40 µL of 1 M CH_3_CO_2_K, 600 µL of MeOH, and 1120 µL of distilled water. The reaction mixture was kept at room temperature for 30 min. The PHE and FLA were spectrophotometrically measured, reading the absorbance of the samples at 765 and 415 nm, respectively, and measured with the calibration curves obtained using increasing amounts of gallic acid (GA) as a standard equivalent (E) for the phenols and quercetin (Q) for the flavonoids. Thus, data were expressed as µg GAE g^−1^ of the FW for phenols, and µg QE g^−1^ of the FW for flavonoids.

### 4.8. Statistical Analyses

All measurements were conducted in triples and the results were reported as the mean ± SD (standard deviation) values. The statistical analyses were performed using PAST 4.10 software (Øyvind Hammer, Oslo, Norway). Shapiro–Wilk tests for verifying the tendency towards a normal distribution were performed for all of the analyzed data; from this, it emerged that the distributions tended towards normal ones (*p* < 0.05) and, therefore, parametric tests (ANOVA) were applied to investigate the data and compare them to one another. Moreover, the parametric Tukey’s post hoc test was also applied. Values of *p* < 0.05 were considered statistically significant. Data comparisons were performed between the spirulina-based treatments and the control group, and also among the different spirulina-based treatment groups.

## 5. Conclusions

In conclusion, the data obtained from the current study made it possible to outline the potential spectrum of applications of biofertilizers based on industrial spirulina waste extract for *L*. *sativa* (cv. Foglia di Quercia Verde) plants grown in an aquaponics system. Among the various administrations of this spray fertilizer, it was possible to identify the 75% dilution as the best one due to its capacity to induce growth of the plant tissues and accumulation of nutraceuticals. As the use of biofertilizers in aquaponics systems is still an area for investigation, the results obtained here are encouraging and lay the foundations for further future analyses. By doing so, some plant foods will possibly be produced with limited environmental impact, but with high nutritional and healthy properties at the same time. It should be emphasized that the total amount of the spirulina biofertilizer used in this study, considering all of the tested dilutions, was approximately half a liter on 60 experimental plants; this is another fact that should not be underestimated because just a small amount of the proposed biofertilizer is needed to induce the described effects using an aquaponics system. Thus, this study has provided a sustainable alternative in favor of a circular economy of resources, offering new ideas for the application of biofertilizers made with waste products which are able to compensate for the lack of nutrients that inevitably occurs in aquaponics cultivation systems.

## Figures and Tables

**Figure 1 plants-12-04030-f001:**
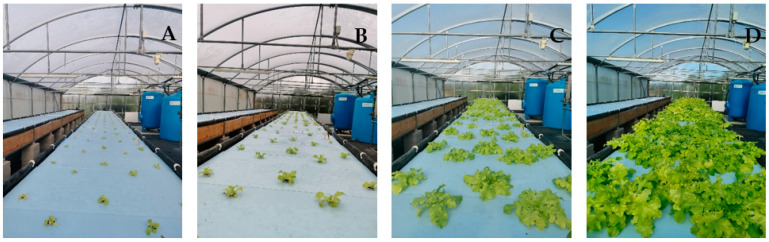
Growth stages of *L. sativa* (cv. Foglia di Quercia Verde) plants in the aquaponics system. From left to right: 30 (panel (**A**)), 45 (panel (**B**)), 55 (panel (**C**)), and 65 (panel (**D**)) days after sowing.

**Table 1 plants-12-04030-t001:** Size measurements relative to the rosette diameter and root length. The units of measurement for each parameter are indicated in brackets.

Treatment	Rosette Diameter (mm)	Root Length (mm)
CT	429.36 ± 37.35 a	930.72 ± 260.82 a
25%	568.66 ± 51.28 b	929.52 ± 198.72 a
50%	584.92 ± 50.85 b	973.65 ± 190.17 a
75%	694.54 ± 40.54 c	946.04 ± 151.96 a
100%	595.96 ± 76.31 b	1046.92 ± 185.88 a

Results are represented by their mean ± SD values. Different letters in each column indicate significant differences (*p* < 0.05).

**Table 2 plants-12-04030-t002:** Measurements of rosette FW, root FW, total FW, and FW/DW ratio. The units of measurement for each parameter are indicated in brackets.

Treatment	Rosette FW (g)	Root FW (g)	Total FW (g)	FW/DW
CT	70.68 ± 29.30 a	14.35 ± 8.41 a	85.03 ± 37.35 a	19.36 ± 0.46 ab
25%	93.01 ± 20.85 ab	14.91 ± 2.75 a	107.92 ± 20.02 ab	18.66 ± 1.28 ab
50%	101 ± 16.94 bc	15.95 ± 4.04 a	116.95 ± 19.60 bc	17.58 ± 0.32 a
75%	128.62 ± 18.64 c	16.63 ± 2.96 a	145.26 ± 21.19 c	20.00 ± 0.99 b
100%	108.33 ± 31.01 bc	21.22 ± 16.23 a	129.55 ± 36.32 bc	20.60 ± 0.99 b

Results are represented by their mean ± SD values. Different letters in each column indicate significant differences (*p* < 0.05).

**Table 3 plants-12-04030-t003:** Sugars and other soluble solids, photosynthetic pigments, and secondary metabolites data: the soluble solid content (SSC), and levels of chlorophyll *a* (CHL *a*), chlorophyll *b* (CHL *b*), carotenoids (CAR), phenols (PHE), and flavonoids (FLA). The units of measurement for each parameter are indicated in brackets.

Treatment	SSC (Brix)	CHL *a* (µg g^−1^ FW)	CHL *b* (µg g^−1^ FW)	CAR (µg g^−1^ FW)	PHE (µg GAE g^−1^ FW)	FLA (µg QE g^−1^ FW)
CT	5.63 ± 1.00 a	36.78 ± 3.73 ab	3.13 ± 2.17 a	17.90 ± 2.15 a	214.44 ± 9.96 a	26.58 ± 2.46 a
25%	8.17 ± 0.76 abc	38.93 ± 2.17 ab	2.33 ± 0.97 a	18.22 ± 1.14 a	399.94 ± 32.85 b	25.91 ± 5.44 a
50%	7.8 ± 1.20 ab	45.76 ± 4.15 a	13.49 ± 3.48 b	23.84 ± 1.72 b	432.18 ± 23.83 b	23.29 ± 1.31 ab
75%	10.83 ± 0.78 c	45.73 ± 3.15 a	13.33 ± 1.63 b	24.41 ± 0.94 b	452.34 ± 24.71 b	18.89 ± 0.58 b
100%	9.67 ± 0.54 bc	36.45 ± 2.64 b	10.17 ± 3.39 ab	19.39 ± 1.36 ab	149.30 ± 14.00 a	22.68 ± 1.34 ab

Results are represented by their mean ± SD values. Different letters in each column indicate significant differences (*p* < 0.05).

**Table 4 plants-12-04030-t004:** Aquaponics system parameters. Specifically: the temperature of the water entering the cultivation tanks (T1), temperature of the water leaving the cultivation tanks (T2), pH, dissolved oxygen (DO), electric conductivity (EC), pressure at the filter level (PRES), and irradiance (IR) above; and the maximum temperature (Tmax), minimum temperature (Tmin), maximum relative humidity (RHmax), and minimum relative humidity (RHmin) below. The units of measurement for each parameter are indicated in brackets.

T1 (°C)	T2 (°C)	pH	DO (mg L^−1^)	EC (S cm^−1^)	PRES (bar)	IR (W sqm^−1^)
23.49 ± 1.41	23.69 ± 1.64	8.6 ± 0.14	8.88 ± 0.95	0.71 ± 0.18	1.08 ± 0.70	371.24 ± 125.29
Tmax (°C)	Tmin (°C)	RHmax (%)	RHmin (%)			
25.48 ± 3.91	9.73 ± 2.90	89.82 ± 2.32	49.35 ± 11.80			

Results are represented by their mean ± SD values.

**Table 5 plants-12-04030-t005:** Concentration of macro- and micronutrients in the water circulating through the aquaponics system and in the algal supernatant section. For each parameter, the relative analytical method is reported.

Target	Units of Measurement	Aquaponics Water	Algal Supernatant	Method
Nitrogen (as NO_3_^−^-N)	mg L^−1^	11.53 ± 5.24	285.00 ± 7.07	Chromotropic acid
Potassium	mg L^−1^	83.33 ± 7.64	250.00 ± 0.00	Adaptation of the turbidimetric tetraphenylborate
Phosphorus	mg L^−1^	3.07 ± 1.27	45.00 ± 14.14	Adaptation of EPA 365.2 and ascorbic acid 4500-PE
Iron	mg L^−1^	0.19 ± 0.09	12.55 ± 2.33	Adaptation of TPTZ
Copper	µg L^−1^	27.67 ± 25.48	0.00 ± 0.00	Adaptation of EPA
Manganese	µg L^−1^	16.00 ± 5.29	1100.00 ± 282.84	Adaptation of PAN
Magnesium	mg L^−1^	19.00 ± 4.58	250.00 ± 70.71	Adaptation of calmagite
Sulfur (as SO_4_^2−^)	mg L^−1^	40.00 ± 3.46	150.00 ± 70.71	Precipitation with barium chloride crystals
Molybdenum	mg L^−1^	0.23 ± 0.15	80.00 ± 14.14	Adaptation of mercaptoacetic acid
Chlorine (as Cl^−^)	mg L^−1^	22.00 ± 2.65	780.00 ± 169.71	Adaptation of mercury (II) thiocyanate
Zinc	mg L^−1^	0.09 ± 0.06	1.5 ± 2.12	Adaptation of zinc from the standard methods for the examination of water and wastewater
Calcium	mg L^−1^	77.00 ± 33.60	910.00 ± 579.83	Adaptation of oxalate

Results were reported as the mean ± standard deviation (SD) values. The abbreviation in the methods column are: Eicosapentaenoic Acid (EPA), 2,4,6-Tripyridyl-S-Triazine (TPTZ), 1-(2-Pyridylazo)-2-Naphthol (PAN).

## Data Availability

Data are contained within the article.

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
