# Peer review of "Sustainability in Aquaponics: Industrial Spirulina Waste as a Biofertilizer for Lactuca sativa L. Plants"

_plants, 2023, doi:10.3390/plants12234030_

Round 1

Reviewer 1 Report

Comments and Suggestions for Authors

-       The introduction does not contain any news that would introduce the reader to the manuscript, but much of the information has already been mentioned in numerous works on aquaponics cultivation

-       Please avoid specifying words from the title as keywords

-       The discussion should be refined. There are a large number of papers that can be used to compare the results obtained in the reasearch (especially lettuce grown aquaponically)

-       The abiotic parameters during lettuce cultivation are not shown. Have you measured pH, EC, and the amount of dissolved oxygen available to the plants during cultivation?

-        The abiotic parameters during lettuce cultivation are not shown. You indicate that you measured pH, EC, and the amount of dissolved oxygen available to the plants during cultivation (line 191). It would be good to list them as this would give a better picture of the growing conditions of the lettuce plants

Reviewer 2 Report

Comments and Suggestions for Authors

The objective of this manuscript "plants-2706937” was demonstrate the application of industrial spirulina waste as foliar biofertilizer in aquaponics. This manuscript seems to present quite relevant information in the current field. The findings are interesting and meaningful.

Comment:

1.            Reference 7 is missing. I guess it is the one in the previous line: Goddek, S.; Joyce, A.; Kotzen, B.; Burnell, G. M., Eds.; Springer Nature: Cham, Switzerland, 2019; pp. 3–187.

2.            The authors' contribution is missing.

3.            Why did it take 3 years to publish the results of two research projects that ended in 2020?

Reviewer 3 Report

Comments and Suggestions for Authors

See PDF
